# Role of Cellular Receptors in the Innate Immune System of Crustaceans in Response to White Spot Syndrome Virus

**DOI:** 10.3390/v14040743

**Published:** 2022-04-01

**Authors:** Ngoc Tuan Tran, Huifen Liang, Ming Zhang, Md. Akibul Hasan Bakky, Yueling Zhang, Shengkang Li

**Affiliations:** 1Guangdong Provincial Key Laboratory of Marine Biology, Shantou University, Shantou 515063, China; tranntts@gmail.com (N.T.T.); 21huifenliang@stu.edu.cn (H.L.); 14mzhang@stu.edu.cn (M.Z.); 19mahbakky@stu.edu.cn (M.A.H.B.); zhangyl@stu.edu.cn (Y.Z.); 2Institute of Marine Sciences, Shantou University, Shantou 515063, China

**Keywords:** shrimp, crab, receptor, innate immunity, antivirus, WSSV

## Abstract

Innate immunity is the only defense system for resistance against infections in crustaceans. In crustaceans, white spot diseases caused by white spot syndrome virus (WSSV) are a serious viral disease with high accumulative mortality after infection. Attachment and entry into cells have been known to be two initial and important steps in viral infection. However, systematic information about the mechanisms related to WSSV infection in crustaceans is still limited. Previous studies have reported that cellular receptors are important in the innate immune system and are responsible for the recognition of foreign microorganisms and in the stimulation of the immune responses during infections. In this review, we summarize the current understanding of the functions of cellular receptors, including Toll, C-type lectin, scavenger receptor, β-integrin, polymeric immunoglobulin receptor, laminin receptor, globular C1q receptor, lipopolysaccharide-and β-1,3-glucan-binding protein, chitin-binding protein, Ras-associated binding, and Down syndrome cell adhesion molecule in the innate immune defense of crustaceans, especially shrimp and crabs, in response to WSSV infection. The results of this study provide information on the interaction between viruses and hosts during infections, which is important in the development of preventative strategies and antiviral targets in cultured aquatic animals.

## 1. Introduction

In crustaceans, innate immunity is known to be the only defense system for resistance against infections [1]. The innate immunity in invertebrates, especially shrimps and crabs, is generally based on humoral responses through the processes of melanization, coagulation, and production of antimicrobial peptides (AMPs) and on cellular responses via encapsulation, phagocytosis, and autophagy [2,3,4]. Cellular receptors (including pattern recognition receptors—PRRs) play dominant roles in recognizing the presence of foreign microorganisms and, in turn, in stimulating the immune responses to respond to infections [5]. Disease outbreak induced by pathogenic microorganisms is one of the important challenges that directly affect the health status of cultured animals. Previous studies have provided clarification of white spot syndrome virus (WSSV), which is one of the most dangerous pathogens causing white spot diseases (WSD) in various cultured crustaceans, such as shrimps, crabs, crayfish, and lobsters [6,7,8,9,10,11,12,13,14]. It has been shown that WSD is a serious viral disease resulting in high cumulative mortality (up to 100%) of cultured shrimps after 2–10 days of infection [15]. Although studies focusing on the infection of WSSV in crustaceans have been widely carried out, information related to the mechanism of infection is still vague. Thus, the mechanisms associated with innate immunity in response to WSSV infection need to be elucidated.

Previous studies reported that attachment onto the surface of cells and entry into cells are two important steps in viral infection cycles [16]. Viral envelope proteins have been known to be crucially important in the entry, assembly, and budding of the virus in the host’s cells [17]. In the case of WSSV, a total of 30 structural proteins have been previously identified [17]. In a recent study, 58 structural proteins (nucleocapsid: 9 and envelope: 32) have been determined in WSSV virions, with the main members being VP19, VP24, VP26 and VP28 [18,19]. The proteins VP19 and VP28 are reported to associate with the envelope, whereas VP24 and VP26 are located at the nucleocapsid [18]. Interestingly, the studies have reported that these viral proteins (such as VP24, VP26 and VP28) are not only responsible for the virion morphogenesis [17] but also for infection of WSSV in several crustaceans [5,20,21,22]. For example, in black tiger shrimp (*Penaeus monodon*), the viral proteins, VP24, VP110, VP53A, VP53B, VP337, VP32, VP124, VP41A, VP51B, VP60A and VP39B, can interact with the host’s cell surface chitin-binding protein (*Pm*CBP) [23]. Moreover, VP110, VP36A, VP36B and VP31 with the presence of an Arg-Gly-Asp (RGD) motif in the genome are able to bind to receptors on the host’s cell surface during the initial stages of infection [24,25]. In the case of viral infection, it has been previously reported that there are specific membrane components as receptors on the host cell surface responsible for the binding of the virus to the envelope proteins and for facilitating their penetration. In crustaceans, the components serving as receptors, such as Toll, C-type lectin (CTL), scavenger receptor (SR), β-integrin, polymeric immunoglobulin receptor (pIgR), laminin receptor, globular C1q receptor (gC1qR), lipopolysaccharide (LPS)-and β-1,3-glucan-binding protein (LGBP), chitin-binding protein (CBP), Ras-associated binding (Rab), and Down syndrome cell adhesion molecule (Dscam), have been reported to be related to the interaction with WSSV during the infection.

This review discusses the current knowledge and recent progress regarding the functions of the cellular receptors participating in the innate immune defense of crustaceans, especially shrimp and crabs, in response to WSSV infection. The results of this study will provide a better understanding of the process of infection and the virus–host interplay during viral infection and play a crucially important role in the development of reasonable strategies used to prevent viral infection in cultured crustaceans.

## 2. Toll Receptors

Toll/Toll-like receptors (TLRs) play important roles in innate immune responses in both invertebrates and vertebrates [26]. Tolls have been identified and shown to play a central role in innate immunity in several crustaceans, such as kuruma shrimp (*Marsupenaeus japonicus*) [27], black tiger shrimp (*P. monodon*) [28,29], red swamp crayfish (*Procambarus clarkia*) [30,31], white leg shrimp (*Litopenaeus vannamei*) [32,33,34], mud crab (*Scylla paramamosain*) [26,35,36]. Structurally, the predicted protein of Chinese white shrimp (*Fenneropenaeus chinensis*) Toll (*FcToll*) is characterized as containing an extracellular domain with a potential signal peptide, 16 leucine-rich repeats (LRR), two LRR-C-terminal (LRR-CT) motifs, and two LRR-N-terminal (LRR-NT) motifs, followed by a transmembrane segment and a cytoplasmic Toll/Interleukin-1R (TIR) domain [37]. In another study, the *SpToll2* has also been cloned from mud crab. The results showed that the *SpToll2* is composed of six extracellular leucine-rich repeat (LRR) domains, a transmembrane domain, and an intracellular Toll/IL-1 receptor (TIR) domain [35]. In crustaceans, Tolls (such as *SpToll*) can be found to distribute in several tissues, such as the heart, gill, hepatopancreas, stomach, intestine, muscle, eyestalk, and hemocytes [26]. Previous studies confirmed that Tolls are crucially important in the crustacean’s innate immunity responsible for the recognition of both bacterial and viral pathogens [26,35,37]. The findings revealed the protective role of Tolls not only in eliminating invading pathogens but also in triggering the signaling pathways involved in the innate immune response. Under stimulation with WSSV, the association of Tolls with the defense system of crustaceans against pathogens has been clearly confirmed. Obviously, the significant induction of *PmToll* (from black tiger shrimp) [29], *MjToll* (from kuruma shrimp) [27], *SpToll* (from mud crab) [26], *SsToll* (from *Scylla serrata*) [38], *SpToll1* and *SpToll2* (from mud crab) [35], and *MrToll* (from giant freshwater prawn- *Macrobrachium rosenbergii*) [39] has been recorded during the challenge with WSSV. In another study, Qiu et al. [40] reported the contribution of *Toll2* in WSSV infection in white leg shrimp. The expression of the *Toll2* receptor was upregulated upon the stimulation of WSSV, but the silencing of *Toll2* significantly increased the survival rate of WSSV-infected shrimp and decreased the viral load in shrimp tissues [40]. The data suggest that Tolls are responsible for the recognition of viral pathogens and may be important in the interaction between host and WSSV. However, some cases, such as *FcToll* (from Chinese white shrimp) [37] or *PmToll* (from black tiger shrimp) [28], were downregulated at early time points of the WSSV challenge or unchanged during a 24 h post-challenge. Collectively, the association of Tolls with the immune response to WSSV may be different from species and infected time points, which need further investigations.

The Toll pathway plays a key role in the response to invading pathogens that appeared outside of the cell and in intracellular endosomes and lysosomes by regulating several related genes [2,41]. In a recent study in white leg shrimp, among nine Tolls (*Toll1-9*) identified, *Toll2*, *Toll3* and *Toll4* were demonstrated to be crucially important in the infection and replication of WSSV through the application of knockdown assays, in vivo [34,40,42]. For example, a previous study showed the role of white leg shrimp *Toll3* (*LvToll3*) in the immune response through the activation of the interferon regulatory factor (IRF)/Vago/JAK-STAT cascade against WSSV infection [42]. The silencing of *LvToll3* decreased the survival rate of WSSV-infected shrimp and downregulated the expression of IRF (at the protein level) [42]. The silencing of *Toll4* showed a significant increase in the viral replication and mortality of shrimps after WSSV challenge, indicating the association of *Toll4* as a major antiviral factor responsible for resistance to WSSV infection in shrimp [34]. The study also confirmed that *Toll4* is a receptor that recognizes WSSV and plays an important role in viral inactivation through the Toll4/Dorsal/AMPs cascade. After recognizing WSSV, *Toll4* stimulates the translocation of *Dorsal* from the cytoplasm to the nucleus and phosphorylation on Ser342; the activated *Dorsal* triggers the expression of specific AMPs (such as anti-lipopolysaccharide factors (ALFs) and lysozymes). and the Toll4-Dorsal-drived AMPs bind to the components of the viral surface, thereby resulting in the inactivation of WSSV [34]. In another study, the existence of the TLR/MyD88/Tube/Pelle/TRAF6/NF-κB pathway has been proven in white leg shrimp [43]. When the shrimp were infected with WSSV, the viral protein (WSSV449), which is structurally similar to the host Tube, can bind to the Death domain of shrimp MyD88, Tube, or Pelle as well as activate the NF-κB signaling pathway by increasing the promoter activities of AMPs [43]. The activation of *NF-κB* by *WSSV449* might be related to the induction of an anti-apoptotic response in the hosts. Furthermore, in this study, *LvPelle* was found to be located in the cytoplasm of cells, which can be induced after WSSV infection and interacted with its downstream component, *LvTRAF6* [43]. The involvement of *LvTRAF6* in the antiviral activity is shown when it was upregulated in the gills and hepatopancreas of shrimps after challenge with WSSV [44]. The results of this study also confirmed the role of *LvTRAF6* (as an adaptor protein in the Toll signaling pathway) in the regulation of AMP gene expression, showing that the activities of *P. monodon Penaeidin*, *L. vannamei Penaeidin4*, *Drosophila Attacin A* and *Drosomycin* increased with the expression of *LvTRAF6* [44]. Similar results were obtained in the case of black tiger shrimp infected with WSSV, where the expression of important components of the Toll pathway, including *PmToll*, *PmMyD88* and *PmTRAF6*, was upregulated at all time points of the infection [29]. Additionally, in mud crab, the identification and characterization of immune function of *SpTube* and *SpPelle* were carried out [45]. The results showed that *SpTube* was downregulated, while *SpPelle* was upregulated in the hemocytes upon WSSV challenge. The binding of *SpTube* to both *SpMyD88* and *SpPelle* indicated the association of these components in the formation of a trimeric complex (MyD88-Tube-Pelle) in the mud crab innate immunity in response to invading pathogens.

*IKKα/β* and *TBK1/IKK**ɛ* are key regulators of the NF-κB signaling pathway and play important roles as a point of convergence for the signal transduction pathways relating to NF-κB activation [46]. After challenges with WSSV, *LvIKKβ* and *LvIKKε* could trigger the activation of downstream signaling molecules and activate the NF-κB pathway, indicating the participation of these components in the IKK-NF-κB signaling pathway. RNAi knockdown of *LvIKKβ* and *LvIKKε* significantly reduced the expression of genes related to the NF-κB signaling pathway, such as *LvPEN2*, *LvPEN3*, *LvPEN4*, *Lvlysozyme*, *Lvcrustin1* and *Lvcrustin2*, as well as increased the survival rates of shrimp after WSSV infection [46]. The increased resistance to WSSV in *LvIKKβ*-or *LvIKKε*-silenced shrimp is possibly the requirement of activation of the IKK-NF-κB signaling pathway for WSSV infection [46]. Moreover, the results of this study confirmed the involvement of *LvIKKβ* and *LvIKKε* in WSSV infection, where the WSSV genes (including *WSSV051*, *WSSV059*, *WSSV069*, *WSSV083*, *WSSV090*, *WSSV107*, *WSSV244*, *WSSV249*, *WSSV303*, *WSSV371* and *WSSV445*) were activated by the overexpression of both *LvIKKβ* and *LvIKKε* [46]. Therefore, the obtained results suggested that *Tolls*, *MyD88*, *Tube*, *Pelle*, *TRAF6*, *IKKs* and *NF-κB* are key components in the Toll signaling pathway responsible for the regulation of antiviral responses in crustaceans. Additionally, the data supported that Tolls may act as PRRs to directly recognize the envelope proteins of WSSV, which in turn stimulate activation of the Toll-MyD88-Tube/Pelle-TRAF6 pathway.

In *Drosophila*, the Toll cannot directly bind to the pathogens, which need an endogenous protein-ligand (Spätzle) for activation and signaling [47]. In the previous study in mud crab challenged with Gram-negative bacterium (*Vibrio parahaemolyticus*), the Toll-associated downstream signaling molecules (such as *Spätzle*, *Toll*, *Cactus*, and *Dorsal*) were upregulated in the hemocytes of mud crab [48]. In the case of WSSV infection, in kuruma shrimp (*M. japonicus*), the Späztle-Toll-Dorsal-ALF signaling pathway was reported to be important in antiviral immunity against invasion [49]. The results showed that *Späztle*, *Toll1*, *Toll4*, *Toll6*, *Toll7* and *Dorsal* are significantly upregulated in WSSV-infected shrimp compared to the control. Silencing of *Späztle*, *Toll1*, *Toll4*, *Toll6*, *Toll7* or *Dorsal* significantly increased the number of WSSV copies or decreased the expression of ALFs [49]. The data provided evidences for the role of Tolls in the indirect interaction with WSSV through *Spätzle* to trigger the activation of the Späztle-Toll-Dorsal-AMPs signaling pathway, which is crucially important in WSSV infection in crustaceans.

## 3. C-Type Lectins

Lectins are typically structured by a carbohydrate recognition domain (CRD) containing 110–130 amino acids and are responsible for binding to sugars [50]. In crustaceans, several groups of lectins, including C-type, F-type, I-type, L-type, M-type, P-type, R-type, chitinase-like lectins, ficolins, calnexin, galectins, and intelectins, have been identified and characterized [50,51]. Functionally, it has been reported that intracellular lectins are related to protein trafficking and sorting, whereas extracellular lectins are involved in pathogen recognition and cell signaling [51]. C-type lectins (CTLs, denoted as calcium-dependent carbohydrate-binding proteins) contain one or more characteristic C-type lectin-like domains and can recognize a wide range of ligands, including glycans, proteins, lipids, and inorganic compounds [2,52]. In crustaceans, CTL has been found in several tissues, for example, *SpCTL-B* is expressed in the brain, muscle, subcuticular epidermis, gills, hepatopancreas, intestines, heart, and hemocytes of mud crab [53]. *SpCTL-B* has been defined as a mannose-binding lectin by containing the EPD motif in its CRD, which is a major pattern-recognition molecule and is capable of binding pathogens. As a PRR in the immune response of mud crab towards the infections, *SpCTL-B* was found to be up-regulated in the hemocytes and hepatopancreas after challenge with *V. parahaemolyticus*, LPS, polyI:C, or WSSV [53]. These data suggest that CTLs may contribute to the immune system in response to WSSV infection.

In crustaceans, after WSSV challenge, the expression of CTLs is greatly affected. The expression patterns of kuruma shrimp CTL after WSSV infection were analyzed through the transcriptomic profile and RT-qPCR. The results showed that WSSV can stimulate the significant upregulation of shrimp CTL [54]. Binding to invaders is an initial and important step for the recognition of PRRs in the innate immune system. Previous studies found that CTLs can bind to the WSSV envelope proteins, which are important for virus entry, assembly, and budding to host cells. In kuruma shrimp, the two lectins *Mj*LecB and *Mj*LecC showed the ability to bind to VP28, whereas *Mj*LecA binds to VP26 [20] and *Mj*CTL binds to VP19, VP24, VP26, and VP28 [21]. Furthermore, in banana shrimp (*Fenneropenaeus merguiensis*), the expression of *FmLC6* increased, peaking at 12 h post-injection, and decreased afterward under challenge with WSSV [55]. The *Fm*LC6 can directly bind to VP15, VP39A, and VP28 with different affinities [55]. In Chinese white shrimp, a new C-type lectin (*Fc*Lec3) was identified and found to be upregulated by WSSV challenge [22]. *Fc*Lec3 has been characterized as a receptor located on the surface of shrimp F cells in hepatopancreas that is able to recognize VP28 using pull-down assay [22]. The binding between lectin and viral envelope proteins may contribute to the prevention of viral entry into shrimp cells, which may be important in prolonging the survival of WSSV-infected shrimp.

In white leg shrimp, *LvCTL1* has been shown to be able to protect shrimp from WSSV infection and to prolong the survival of WSSV-infected shrimps [19] or the downregulation of *LvCTLU* [56], and silencing of *LvCTL5* increased the mortality of shrimp after infection with WSSV [57]. Additionally, silencing of a low-density lipoprotein receptor class A domain-containing C-type lectin (*LvLdlrCTL*) showed an increase in the mortality and a decrease in the viral load in the muscle of white leg shrimp after challenge with WSSV, indicating that *LvLdlrCTL* could facilitate the viral infection [58]. In another study, the results have shown an upregulation of *LvCTLU* in the gills, hepatopancreas, and intestine at different time points of WSSV infection [56]. Furthermore, Zhao et al. [19] found that the binding of recombinant *Lv*CTL1 and WSSV envelope proteins (including VP95, VP28, VP26, VP24, VP19 and VP14) significantly decreased the mortality of shrimps after challenge with WSSV and protected shrimp hemocytes against WSSV infection. Additionally, the *LvCTL 4.2* showed its binding ability to the WSSV virion and the silencing of *LvCTL 4.2* reduced the replication of WSSV, but pretreatment of WSSV with r*Lv*CTL 4.2 allowed for viral replication in vivo [59].

Collectively, the results of these studies indicated that CTLs serve as PRRs to recognize WSSV by binding to the viral envelope proteins, facilitating immune defense by helping the hosts eliminate invading viruses [19,55]. However, the mechanisms involved in the viral infection of CTLs should be further investigated.

Interestingly, in a previous study, the kuruma shrimp CTL (*MjCC-CL*) can recognize the glycans of a pathogen surface (via C-type carbohydrate recognition domain) and bind to *Dome* (via coiled-coil domain) to trigger the activation of the JAK/STAT pathway and in turn to stimulate the phosphorylation and translocation of *STAT*, thereby resulting in an upregulation of AMPs (such as *ALF-C1*, *ALF-C2*, *ALF-D1*, *CruΙ-1* and *CruΙ-5*) [60]. Similar to this observation, the previous results have found that silencing of *LvLdlrCTL* leads to the down- or upregulation of many immune effector genes, including *Dorsal*, *Relish*, *STAT*, and *IRF* [58]. This suggested the role of *LvLdlrCTL* in upstream regulatory pathways involved in these genes (such as NF-κB and JAK/STAT pathways) [58]. Obviously, several components associating with the JAK/STAT pathway, such as *JAK*, *STAT*, *SOCS* and *SpDicer2*, have been reported to attend to the immune system of crustaceans in the defense against WSSV [61,62,63,64,65,66,67]. Additionally, the existence of the IRF-Vago-JAK/STAT regulatory axis in response to WSSV infection has been demonstrated in white leg shrimp [68]. Previous findings have given evidence for a regulatory role of CTL in the association with the activation of many signaling pathways in response to WSSV infection in crustaceans. However, the mechanisms that explain the antiviral activity of CTL when interacting with other pathways in crustaceans under the WSSV infection need to be studied in-depth in further investigations.

## 4. Scavenger Receptors

Scavenger receptors (SR) are cell surface receptors that interact with various ligands and promote the elimination of degraded or harmful substances (including non-self or altered self-targets) and show their function via endocytosis, phagocytosis, adhesion, and signaling [69]. SRs play a key role in the innate immunity of both vertebrates and invertebrates by recognizing pathogen-associated molecular patterns (PAMPs) and in the pathogenesis of diseases through interactions with damage-associated molecular patterns [70,71]. In crustaceans, one of the class B scavenger receptors, Croquemort, designated as *MjSCRBQ*, was obtained from a cDNA library of the lymphoid organs of kuruma shrimp [72]. *MjSCRBQ* has been characterized to contain transmembrane domains and a CD36 domain. *MjSCRBQ* was found to be constitutively expressed in multiple tissues, with a high level in the brain. Under WSSV stimulation, the expression of *MjSCRBQ* was upregulated in lymphoid organs of kuruma shrimp at 72 and 120 h post-infection [72]. Another study also proved that the class C scavenger receptor (*MjSRC*) obtained from kuruma shrimp is also expressed in several tissues (including hemocytes, heart, hepatopancreas, gills, stomach, and intestine) [73]. *MjSRC* was found to be upregulated in hemocytes from 6–72 h of WSSV injection [73]. *MjSRC* confers a WSSV phagocytotic receptor participating in the hemocyte phagocytosis during WSSV infection. In this study, the authors have described the protective role of *MjSRC* in the hemocytes of kuruma shrimp against WSSV infection [73]. After WSSV infection, *MjSRC* is stimulated and oligomerized to a trimer that recognizes WSSV by the interaction between its extracellular domain (MAM-domain in meprin, A5, receptor protein tyrosine phosphatase mu) with a viral envelope protein (VP19) and initiates antiviral responses in shrimp. *MjSRC* also forms homo-trimers from the CCP (complement control protein) and MAM domains and is subsequently internalized from the plasma membrane into the cytoplasm after the infection. Moreover, the intracellular region of *MjSRC* recruits and interacts with *Mjβ-arrestin2*, activating the phagocytosis of WSSV by hemocytes via the *Mj*SRC-arrestin-clathrin pathway. Finally, WSSV is degraded in the phagosomes by fusing with a lysosome, thereby effectively inhibiting viral infection in shrimp [73].

In mud crab, the class B scavenger receptor (*Sp-SRB*) was obtained and found to distribute in several tissues (including hemocytes, mid-intestine, hepatopancreas, muscle, stomach, subcuticular epidermis, gills and heart) [74]. *Sp-SRB* was induced in both hemocytes and hepatopancreas of mud crabs after challenge with either WSSV, *V. parahaemolyticus*, LPS, or PolyI:C [74]. In this study, the results showed that *Sp-SRB* is related to the promotion of bacteria clearance by enhancing phagocytosis in the case of bacterial infection only [74], but the information requires confirmation in the case of viral infection in further studies. Moreover, during the bacterial infection (with *V. parahaemolyticus* or *S. aureus*), *Sp-SRB* has been shown to be able to cooperate with Tolls to regulate the expression of AMPs in mud crab. Furthermore, silencing of *Sp-SRB* decreased the number of WSSV copies but increased the apoptosis rate in the hemolymph and activity of *caspase-3* in the hepatopancreas of mud crab [74]. This indicated that *Sp-SRB* plays an important role in stimulating apoptosis in mud crab after WSSV infection. In addition, a mud crab class B scavenger receptor (*SpSR-B2*) was found to be expressed in the gills, hepatopancreas, stomach, and connective tissue and to be upregulated in the gills of mud crab after challenge with WSSV or bacterial pathogens (such as *V. parahemolyticus* and *S. aureus*) [5]. This indicated the relationship of *SpSR-B2* with the antiviral and antibacterial immune responses in mud crabs. The results of the study also showed that *SpSR-B2* can specifically bind to VP24 or VP26 but not to VP19 and VP28. Silencing of *SpSR-B2* significantly attenuated the expressions of AMPs (including *Crus1*, *Crus4*, *Crus5*, *lys-i* and *ALF2*) in mud crab, while overexpression of *SpSR-B2* upregulated those of *SpALF2*. The findings of this study suggested that *SpSR-B2* serves as a PRR recognizing WSSV through the binding to viral envelope proteins, which in turn initiates the *Sp*SR-B2-mediated signaling pathway to induce the expression of certain AMPs and subsequently resists the replication of WSSV [5].

Taken together, the findings suggested that scavenger receptors are PRRs that play a crucially important role in the innate immune response of crustaceans in defending WSSV infection. However, the underlying molecular mechanisms by which SRs regulate the response to viral infections (such as participating in the hemocyte phagocytosis) remain unclear.

## 5. β-Integrin

Integrins are cell adhesion receptors that comprise a family of more than 23 noncovalent, heterodimeric complexes consisting of an α and a β subunit, with an extracellular domain, a transmembrane domain, and a small cytoplasmic domain in each subunit [75]. Integrins are reported to be important in the physiological processes, comprising cell migration, development, wound healing, immune system function, and pathogenesis of several diseases [76]. As cellular receptors, integrins are known to support the internalization of viruses into the host cells [77,78]. Surprisingly, integrin has been reported to be a heterodimeric protein including α and β subunits, found in several species such as humans, mouse, chicken, zebrafish, nematode (*Caenorhabditis elegans*), African clawed frog (*Xenopus laevis*), and fruit fly (*Drosophila melanogaster*) [79], but α subunit has not yet been found in crustaceans [80].

In crustaceans, integrins (such as Chinese white shrimp *FcβInt*) were found to be distributed in many tissues, including hemocytes, heart, gills, gonad, intestine, muscle, and hepatopancreas with different levels [80]. In recent, studies have shown that integrin β subunit (β-integrin) is important in the infection of pathogens (including viral infection). In Chinese white shrimp, the studies proved that the expression *FcβInt* is induced in the gills [80] and in different types of hemocyte (such as semi-granular and granular cells) [81] by WSSV infection. The upregulation of β-integrin under stimulation with WSSV could activate the integrin-related signaling pathways that facilitate the propagation of WSSV in hosts [77]. *FcβInt* is important in WSSV infection when the polyclonal antibodies against recombinant extracellular region of *FcβInt* (r*Fc*βInt-ER) are found to have a binding activity to WSSV [80]. This suggested a function as a cellular receptor (or post-attachment receptors or co-receptors) of *FcβInt* for WSSV infection. Moreover, a recent study has determined the binding specificity of *Fc*βInt-ER to WSSV envelope proteins (such as VP31, VP37, VP110 and VP187) [77]. The functional role of *FcβInt* in WSSV infection was demonstrated using integrin-specific antibodies to block the WSSV infection and gene silencing to confirm the suppression of reduced WSSV gene transcripts in the hemocytes of shrimps, in vivo [77].

In the case of WSSV infection in kuruma shrimp, β-integrin has been found to bind to VP187 [82]. Gene silencing of the shrimp integrin inhibited WSSV infection and promoted the survival rate (64%) of shrimp after the challenge with WSSV [82]. The data indicated the role of integrin as a receptor for WSSV infection in shrimp. After the interaction with integrin, WSSV triggers the activation of focal adhesion kinase (FAK), which initiates the regulation of downstream signaling pathways [83]. In kuruma shrimp, the study found that a novel FAK from kuruma shrimp (*MjFAK*) was found upregulated after WSSV challenge, implying its functions in WSSV infection and promotion of cell adhesion at the early stage of infection [84]. The authors have speculated that the virus uses the interaction between viral proteins and integrin to activate the induction of *MjFAK*/phosphorylation and therefore facilitates entry of the virus from the plasma membrane into the cytoplasm as well as release of the viral genome to the nucleus [84]. In another study, Lu et al. [83] have also reported that the mortality of FAK-silenced kuruma shrimp increased after challenge with WSSV. The viral protein kinase WSV083 can inhibit the activity of FAK through the increase in its serine phosphorylation to suppress the tyrosine phosphorylation of *MjFAK*. Additionally, WSV083 can inhibit the activities of cell adhesion and related kinase, enabling the virus to evade the host immune response [83].

Furthermore, the white leg shrimp β-integrin (*Lv*Int) was reported to interact with the WSSV envelope (VP26, VP31, VP37 and VP90) and nucleocapsid proteins (VP136) [85]. The RGD-, YGL- and LDV-related peptides may play important roles as motifs of WSSV proteins in interacting with β-integrin [85]. In red swamp crayfish, the study showed that β-integrin can bind to VP187 and block WSSV infection [82]. Interestingly, the conserved integrin-binding motif (RGD) was also contained in the WSSV envelope proteins (VP110, VP36A, VP36B and VP31) [24,25], which may interact with integrins and play important roles during the initial stages of the virus infection.

Therefore, these results showed that integrins may be receptors for the attachment and entry of WSSV into the host cells, which stimulate the signaling pathways in the immune system of crustaceans in response to WSSV infection. However, the molecular mechanisms relating to the crosstalk of integrin-related signaling pathways and WSSV infection should be considered in details in further investigations.

## 6. Polymeric Immunoglobulin Receptor

Polymeric immunoglobulin receptor (pIgR) is a transmembrane receptor that plays an important role in host defense responsible for transporting antibodies across mucosal epithelial cells [86]. In mammals and avian species, pIgR is expressed on the basolateral membrane of the epithelial cells, where it can trap polymeric immunoglobulin (Ig) molecules and transport Ig through the cell to the mucosal surface [87]. In white leg shrimp, Xiao et al. [88] have found that *Lv*pIgR, acting as a host entry receptor for the virus, is able to interact with WSSV VP24. In kuruma shrimp, the *MjpIgR* (containing a signal peptide, an extracellular domain, including an IG domain and two IG-like domains, a transmembrane region, and an intracellular region) was found expressed in the hemocytes, heart, hepatopancreas, gills, stomach, and intestine [89]. The involvement of *MjpIgR* in WSSV infection has been proven by the increased expression after challenge, the facilitation of WSSV invasion (via gene silencing, blocking, or gene overexpression assays), the interaction with viral envelope protein (VP24), the passage of virus through the cellular membrane barrier (via pIgR-CaM-clathrin-mediated endocytosis pathway), and the viral replication and colocalization (via the classical adaptor protein complex AP-2) [89]. The infection of WSSV to the host cells through pIgR initiates with the stimulation of the binding between *MjpIgR* and WSSV on the induction of *MjpIgR* oligomerization to tetramers, and then, the signal is transferred to the cell cytoplasm. The intracellular domain of *MjpIgR* recruits and binds to calmodulin (*MjCaM*), and the *MjCaM* interacts with clathrin (*Mjclathrin*) and *AP-2* adaptor complex, subsequently resulting in the endocytosis of WSSV into the host’s cells [89]. The results indicate the important role of pIgR in WSSV infection.

## 7. Laminin Receptor

Laminin receptor, which is known as 37 kDa laminin receptor precursor (37-LRP), 67 kDa high-affinity laminin receptor (67-LR), 37 kDa/67 kDa laminin receptor, p40 ribosome-associated protein, and Lamr/p40, has been recognized as a multifunctional protein relating to several biological processes (such as cell adhesion, mobility, and differentiation) [90]. Laminin receptor is known to be a cell surface receptor that can bind to capsid or envelope proteins of viruses (such as infectious myonecrosis virus and yellow head virus) [90]. In shrimp, laminin receptor was identified for the first time from black tiger shrimp, which has a capacity of binding to the capsid proteins (VP1, VP2 and VP3) of Taura syndrome virus [91]. In the case of WSSV infection, black tiger shrimp laminin receptor (*Pm*Lamr) was found to interact with nine viral envelope proteins (including VP16, VP19, VP31, VP38B, VP41A, VP41B, VP52B, VP150 and VP187) based on yeast two-hybrid screening [92]. The interaction between *Pm*Lamr and VP31 can reduce the mortality of shrimp after challenge with WSSV [92]. In a recent study, a *CqLR-like* gene containing a conserved laminin-binding domain was shown to involve in WSSV infection in red claw crayfish (*Cherax quadricarinatus*) [93]. The results of the study showed that CqLR-like can interact with VP28 but not others (VP19, VP24 and VP26) by pull-down assay in HEK293T cells. Silencing of *CqLR-like* gene decreased WSSV entry and replication in hematopoietic tissue cells [93]. The results suggested that the laminin receptor is a cellular receptor interacting with WSSV envelope proteins and mediates the passage of the virus into the host cells; however, the molecular mechanisms remain unclear.

## 8. Globular C1q Receptor

Globular C1q receptor (gC1qR) is a multifunctional and multicompartmental cellular protein and is known as a receptor for extra- and intracellular proteins, and microbial and viral proteins [94,95]. gC1qR binds to complement protein C1q, which induces early defense against viral infection as well as regulates adaptive immune response and the viral invasion through the complement and kinin/kallikrein pathways [96]. In crustaceans, several studies have focused on the functions of gC1qR as a PRR to bind to microorganisms and PAMPs in innate immunity [96,97,98]. The crustacean C1qR is characterized to contain a mitochondrial targeting sequence and a mitochondrial acidic matrix protein (MAM33) domain [96,97,98,99]. In freshwater crayfish (*Pacifastacus leniusculus*), the *PlgC1qR* transcript was detected in several tissues, with a much higher expression in hepatopancreas and heart [99]. gC1qR can promote the infection and can maintain the persistence of the hepatitis virus through binding to the viral core proteins [95]. The previous study showed that WSSV infection increased the expression of giant freshwater prawn *MrgC1qR* (in hepatopancreas) [97] and Chinese white shrimp *FcgC1qR* (in hemocytes) [100] after 6 h post-injection, implying the involvement of gC1qR in immune activities. Under the WSSV infection, gC1qR can bind to the viral envelope proteins, for example, *Pl*gC1qR binds to VP15, VP26 and VP28 [99] and *Mj*gC1qR binds to VP15 [101]. Watthanasurorot et al. [99] conducted both in vitro (through hematopoietic tissue cells) and in vivo experiments (through freshwater crayfish individuals) in order to determine the function of *PlgC1qR* in antiviral activity against WSSV. The results found that the expression of *PlgC1qR* is induced by WSSV infection; the silencing of *PlgC1qR* increased the WSSV replication, and the recombinant *Pl*gC1qR protein reduced the viral replication [99]. Collectively, these data suggested that gC1qR plays an important role as a PRR in the innate immunity of crustaceans following WSSV infection.

## 9. Lipopolysaccharide and β-1,3-Glucan-Binding Protein

Lipopolysaccharide-and β-1,3-glucan-binding protein (LGBP) or Gram-negative binding protein can specifically recognize both LPS existing on the cell surface of Gram-negative bacteria and β-1,3-glucan of fungi [102]. LPS and β-1,3-glucans, under the binding to LGBP, induce degranulation of hemocytes and subsequently promote activation of the prophenoloxidase system, leading to the synthesis of melanin through the oxidation of phenols and the activation of genes for antibacterial effector proteins [103]. In shrimp (*Penaeus stylirostris*), the LGBP gene has been characterized to comprise a putative signal sequence, two potential glycosylation sites, a potential recognition motif for β-1,3-linkage of polysaccharides, two putative cell adhesion sites (Arg-Gly-Asp), and a protein kinase C phosphorylation site [104]. LGBP (such as ridgetail white prawn, *Exopalaemon carinicauda* LGBP-*Ec*LGBP) was found expressed in various tissues, with the highest level in the hepatopancreas [105].

The critical role of LGBP in the immune system by regulating the activation and/or activity of the prophenoloxidase cascade in *P. stylirostris* against WSSV has been previously reported [104]. Moreover, in a study to determine the efficacy of a vaccine against WSSV in black tiger shrimp, Syed and Kwang [106] compared the expression level of the LGBP gene in WSSV-treated and Bac-VP28-treated shrimp at different time points. The results showed that LGBP is upregulated in the WSSV-treated shrimp, whereas downregulated in the Bac-VP28-treated shrimp at 7-and 10 days post-infection. The upregulation of LGBP from ridgetail white prawn and white leg shrimp was observed after challenge with WSSV, indicating the involvement of LGBP in the immune defense against viral infection [102,105]. In white leg shrimp, the importance of *Lv*LGBP in innate immunity was confirmed by gene silencing [102]. The results showed the higher cumulative mortality (100%) in *Lv*LGBP-dsRNA-silenced shrimp compared with the control (40–50%) after 72 h post-WSSV infection [102]. Although previous studies have proven the potential importance of LGBP as a pathogenic recognition protein in the activation of shrimp immune defense against invading pathogens (such as *V. parahaemolyticus*) via the agglutination, binding and enhancing encapsulation, and phenoloxidase activity of the hemocytes [102], this issue remains unclear in the case of infection with WSSV, which merits further investigations.

## 10. Chitin-Binding Protein

Proteins that bind to chitin are termed chitin-binding proteins (CBPs), which have been reported to have antimicrobial activity in crustaceans [107]. CBPs were speculated to be located on the cell surface [23] and characterized to consist of one or more chitin-binding domains [107]. In crustaceans, the CBP gene or CBP homologous gene was detectable in many species, such as white leg shrimp, Japanese swamp shrimp (*Neocaridina denticulate*), giant freshwater prawn, bamboo shrimp (*Atyopsis spinipes*), red swamp crayfish, oriental river prawn (*Macrobrachium nipponense*), and kuruma shrimp [108]. It has been shown that, for example, the black tiger shrimp CBP (*Pm*CBP) contains a conserved chitin-binding Type 2 domain, comprising a 6-cysteine motif and several aromatic residues [109]. *Pm*CBP distributes in several tissues [109]. The previous study using expressed sequence tag approach found that a group of immune-related CBP genes seems to be strongly upregulated after WSSV infection [110]. The results were confirmed by the time-course RT-PCR analysis, showing upregulation of *Pm*CBP at the late stage of WSSV infection [109]. The function of *Pm*CBP in viral infection was determined when it was found to directly interact with WSSV067 (encoding the WSSV envelope protein VP53A) [109] and other envelope proteins (including VP24, VP53A, VP110, VP53B, VP337, VP32, VP124, VP41A, VP51B, VP60A and VP39B) of WSSV [23]. Additionally, the recombinant *Pm*CBP produced by *Escherichia coli* can enhance the survival rate of shrimp after WSSV challenge using in vivo neutralization experiments [23]. In white leg shrimp, CBP has been reported to play an important role in WSSV infection when it showed an ability to interact with the viral envelope proteins, including VP24, VP31, VP32, VP39B, VP56 and VP51B [108,111]. Knockdown of CBP can decrease the cumulative mortality of shrimp after WSSV infection [108]. Another study has proven that the binding of WSSV to chitin through its envelope protein (VP24) is necessary for viral *per os* infection in white leg shrimp [112]. The interaction between VP24 and chitin enables the virus to bind and cross through the chitinous barriers in the digestive tract, which thereby facilitates WSSV interacting with *Pm*CBP via other envelope proteins on the cell surface and mediates virus entry [112]. Moreover, the study found that VP24 contains a chitin-binding domain (CBD) with a peptide (P-VP24_186–200_) that can inhibit both VP24-chitin-binding and WSSV-chitin binding, which can be orally applied to prevent WSSV infection [112].

Cuticle proteins (containing CBDs) and chitin are the main components of the cuticle covering the entire outer surfaces and the gastrointestinal tract of crustaceans, which serve as a protective barrier against damages and infections [113]. During WSSV infection, the expressions of some genes encoding cuticle proteins in shrimp were significantly induced, indicating the involvement of cuticle proteins in WSSV infection of shrimp [10,110,114]. In white leg shrimp, previous studies using transcriptome analysis showed that several CBD-containing proteins respond to WSSV infection [115]. Additionally, the results of RT-qPCR showed the expression of genes encoding cuticle protein (i.e., U19114 and U33927) and thrombospondins (i.e., CL6328 and U22382), suggesting the potentially important role of proteins containing CBDs in response to WSSV infection [115]. Moreover, in recent studies, the cuticular CBPs, including *Lv*AMP13.4, *Lv*DD9A, *Lv*DD9B and *Lv*CPAP1, of white leg shrimp were found to be significantly upregulated (at the mRNA level) after challenge with WSSV [10,113,116]. The study also found that *Lv*CPAP1 and *Lv*AMP13.4 are able to interact with WSSV envelope protein (VP24) via yeast two-hybrid system, dot blot, and pull-down assays [10,113]. The silencing of *Lv*DD9A/B inhibited both WSSV gene expression and genome replication [116], and the silencing of *Lv*CPAP1 and *Lv*AMP13.4 decreased the number of WSSV copies and the mortality of shrimp after WSSV infection [10,113]. These results indicate that the host cuticle proteins are related to the entry of WSSV from the cytoplasm membrane into the cells by their interaction with VP24. The data showed that, although the silencing of *Lv*CPAP1 and *Lv*AMP13.4 shows a positive effect on the immune response of shrimp against WSSV, the results are not completely perfect during the infection period. This suggested that cuticle proteins are one of the most important receptors in the host that respond to WSSV [10]. Additionally, other proteins, such as chondroitin proteoglycan 2, consisting of CBD in their structure, have shown an association with the immune response against WSSV infection. The chondroitin proteoglycan 2 of white leg shrimp (*Lv*CPG2) was upregulated in the lymphoid, hepatopancreas, and intestine of shrimp after challenge with WSSV and was demonstrated to be involved in the facilitation of viral infection [117]. *Lv*CPG2 can interact with VP26 and VP28, indicating that *Lv*CPG2 may be important in WSSV adhesion and penetration of shrimp hemocytes. The differences in the protein–protein interactions between *Lv*CPG2 and other CBD-containing proteins with WSSV envelope proteins suggest a different mechanism among them in facilitating WSSV infection [117].

## 11. Ras-Associated Binding Proteins

The small GTPases, Ras-associated binding (Rab) proteins, are important as regulators in vesicle transport for protein delivery to specific intracellular locations, membrane, budding, and fusion events [118]. In crustaceans, Rab and other small GTPase (Ranbp and Rho) were found in the subtractive library of the WSSV-resistant shrimp, suggesting that these proteins play an essential role in living cells [119]. Additionally, Rab7 has been proven to contribute to the entry of viruses (including WSSV) when it is bound to the major WSSV envelope protein (VP28) [88]. The black tiger shrimp Rab7 (*PmRab7*) was the first to be identified and characterized [120]. *Pm*Rab7 contains four conserved GTP-binding or GTPase regions of the small G protein superfamily and an effector site, which is expressed in many tissues (including hepatopancreas, hemocytes, stomach, lymphoid organ, gills, and heart). *Pm*Rab7 is known to be associated with the WSSV infection in shrimp through its ability to bind to a major viral envelope protein (VP28) and blocking the WSSV infection in shrimp [120]. Silencing of *PmRab7* dramatically inhibited the expression of WSSV envelope protein (VP28) at both mRNA and protein levels [121]. Interestingly, the silencing of *PmRab7* decreased the mortality of shrimp (approximately 45%) after WSSV challenge, indicating that Rab may affect the virus in accumulation in the early endosome. However, the dsRNA viral gene (dsRNA-*rr2*) alone or in combination with dsRNA-*PmRab7* showed the best inhibitory effect on the mortality of shrimp (approximately 5%) after WSSV challenge [122]. Additionally, *Rab9* has been found to be important in antiviral response in Japanese shrimp (*M. japonicas*) against WSSV through regulating autophagy [123]. The results showed that *Rab9* is upregulated in shrimp challenged with WSSV. The silencing of *Rab9* resulted in an increase in WSSV copies and a decrease in autophagy levels. The expression of *Rab9* was induced during the autophagy process. The results of this study showed that *Rab9* silencing promotes autophagy of the host’s cells [123]; however, the relationship between *Rab9* and autophagy has not been clearly understood.

## 12. Down Syndrome Cell Adhesion Molecule

Down syndrome cell adhesion molecule (Dscam) is a large protein (~220 kDa), comprising more Ig domains than most other members of the Ig superfamily [124]. Dscam has been known to be important in neuronal wiring and innate immunity in both insect and crustacean species [125]. In crustaceans, for example, the freshwater crayfish Dscam (*Pl*Dscam) has been characterized as containing ten Ig domains, six fibronectin type 3 domains (FN III), and a transmembrane domain [125]. Dscam is known as a PRR and is associated with the innate immune system of crustaceans, such as bacterial clearance and phagocytosis [125,126]. Under WSSV infection, the previous study has shown that different white leg shrimp Dscam (*LvDscam*) isoforms (such as Ig2 and Ig3 variable regions) tend to be present in shrimp at the different WSSV-infection states [127]. In another study, Chiang et al. [128] found that total *LvDscam* and its forms (tail-less *Lv*Dscam and membrane-bound *LvDscam*) are significantly upregulated in the shrimp hemocytes after WSSV infection. Furthermore, in another study, it has been found that the unique Ig2/Ig3 variants of *Lv*Dscam can interact with the VP28 using both yeast two-hybrid and GST pull-down approaches [129]. Moreover, the silencing of *LvDscam* can decrease specific hemocytic phagocytosis and the mortality of shrimp against WSSV induced by *Bacillus subtilis* spores harboring VP28 delivery (rVP28-bs) [129]. The obtained results indicated the involvement of *LvDscam* in the specific immune response of shrimp against WSSV induced by rVP28-bs. In addition, Li et al. [126] suggested the role of Dscam as an inducible PRR in mud crab, which was found to be induced at different time points in the hemolymph and brain of mud crab after challenge with WSSV. Nevertheless, in the case of WSSV infection in freshwater crayfish, *PlDscam* may not be involved in the WSSV infection when it did not respond to WSSV infection, and the gene silencing of *PlDscam* did not affect WSSV infection or replication [125]. Therefore, there are differences in the functional characterization of Dscams that originated from different host species; thus, further studies to provide a better understanding of the ways that Dscam is related to invertebrate adaptive immunity are necessary.

## 13. Conclusions

Cellular molecules, which have been characterized as important receptors, are the first line of defense in the innate immunity of crustaceans during WSSV infection. The receptors are responsible for the attachment, entry, and internalization of the virus as well as the stimulation of the activation of downstream molecules, which result in the regulation of components attending to the proliferation or inhibition of the virus. However, the potential molecular mechanisms for receptors (alone or in combination with other receptors) and their related signaling pathways in antiviral immunity have remained unknown, which merits further investigations. The processes of attachment and entry, internalization, replication, and release of WSSV in host cells need to be investigated to provide a better understanding of the interactions between WSSV and host cells during infections. Recent work has demonstrated the application of short-chain fatty acid-producing bacteria, and their metabolites (including butyric acid) can confer health benefits in aquatic animals [130,131], but their benefits in improving the immune system of crustaceans in response to WSSV infection have not been shown. Additionally, our previous work showed the involvement of exosomal miRNAs in the anti-WSSV response of mud crabs [132]; however, the crosstalk between exosome and host receptors has not yet been fully understood, which merits further investigations. Interestingly, in a recent study, HUWE1 (HECT, UBA and WWE domain-containing E3 ubiquitin-protein ligase 1) and TRAF6 have been shown to be important in regulating the WSSV replication by promoting p53 ubiquitination and by affecting reactive oxygen species and apoptosis during WSSV infection in the mud crab [133]. However, in this present study, the cell-surface receptors related to the attachment and entry of virus have not been studied, which are worthy of further exploration.

## Data Availability

Not applicable.

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
