# Peer review of "Role of Cellular Receptors in the Innate Immune System of Crustaceans in Response to White Spot Syndrome Virus"

_viruses, 2022, doi:10.3390/v14040743_

Round 1
Reviewer 1 Report
Reviewers' Comments to Authors:
The review manuscript entitled “Role of Cell-Surface Receptors in the Innate Immune System of 2 Crustaceans in Response to White Spot Syndrome Virus” by Tran et al., describing a vigorously scientific attempts to narrate functional roles and expression analysis of pattern recognition receptor genes crucial taking part in response to white spot syndrome virus, which is a major pathogen that causes severe diseases (white spot disease; WSD) in several crustaceans in decapod invertebrates, especially for prawn, shrimp and crab.
All in all, the review manuscript is well-written and well describe critical information useful for understanding immune pathways in crustaceans used to respond to WSSV especially for host-pathogen interaction via immune receptor molecules of crustaceans. However, it still remains some minor concerns that the authors should revise and provide some correction to improve the quality of the review manuscript, these included;
Abstract
Line 11. Correct “white spot syndrome” to “white spot diseases (WSD)” and in every place.
Introduction
Line 29. To increase the better flow of the sentence, please correct “in invertebrates” to “in crustaceans”
Line 33-35. The logic of the following is awkward; “Cell-surface receptors (including pattern recognition receptors-PRRs) play an important role in the innate immune…”, since many cell-surface receptors do not play roles in innate immune system…”, please properly revise.
Line 76. Please carefully consider about “in cultured aquatic animals”, more specific terms should replace “aquatic animals”, since only crustaceans are normally affected by WSSV.
Basic interaction of PAMPs and PRRs should be properly raised in this part to indicate the major direction of this review.
- Toll receptors
Line 82 and line 103. Please keep consistent either using “white leg shrimp (Litopenaeus vannamei) or Pacific white shrimp (Penaeus vannamei)”, since it is the same organisms.
Line 116. Change a grammatical error “has” to “have”.
Line 159-162. Please recheck the truth of the following sentence; “RNAi knockdown of LvIKKβ and LvIKKε significantly reduced the expression of genes related to the NF-κB signaling pathway, such as LvPEN2, LvPEN3, LvPEN4, Lvlysozyme, Lvcrustin1, and Lvcrustin2, as well as increased the survival rates of shrimp after WSSV infection [46].”. How reduction of such “LvPEN2, LvPEN3, LvPEN4, Lvlysozyme, Lvcrustin1, and Lvcrustin2” increase survival rate of shrimp after WSSV infection??
- C-type lectins
Line 216-217. Please consider the following sentence carefully; “FcLec3 has been characterized to be a receptor located on the surface of shrimp cells that is able to recognize VP28 using pull-down assay [22].”. The authors should specify “shrimp cells”. Does FcLec3 ban be ubiquitously observed in every kind of cells in shrimp??
Line 235. Correct “Collective”.
Line 243. Please be careful indication of “gene or protein” status using Italic forms in all gene status in everywhere else.
Line 252-253. Please correct grammatical errors and direction of meaning of the following sentence; “The evidence for a regulatory role of CTL in the association with the activation of many signaling pathways in response to WSSV infection in crustaceans.”
- Scavenger receptors
Line 260-261. Please carefully consider the following sentence; “SRs play a key role in the innate immunity of both mammals and invertebrates through recognizing pathogen-associated molecular patterns (PAMPs) and in”. To increase the better flow, please change “mammals” to “vertebrates”.
Line 287-289. Please carefully check the following information; “Sp-SRB was induced in both hemocytes and hepatopancreas of mud crabs after challenge with WSSV or other pathogens (including V. parahaemolyticus, LPS, or PolyI:C) [74].”. LPS or POlyI:C is not classified as pathogens.
Line 291. Please correct grammatical errors of relative pronounce “which” and many places throughout the manuscript.
Line 296. Correct “Casepase-3” to “casepase-3”.
- b-intergrin
Line 322. The following information is vague; “Surprisingly, integrin has been reported to be a heterodimeric protein including α and β subunits, but α subunit has not yet been found in 323 crustaceans [79]”. The authors should specify, what kind of animals that heterodimeric intergrin α and β subunits is reported.
- Globular C1q receptor
Line 434. Please check what kind of cells used for this experiment, “ a hematopoietic tissue cell culture or a primary cell-line)”??
- Lipopolysaccharide-and β-1,3-glucan-binding protein
Line 460. Please verify and clarify “the vaccinated group”??
- Chitin-binding protein
Line 495. Please check “chitin” in this part. Does it refer to “chitin or chitin binding protein”??
Line 511. Correct “suggests”.
- Conclusion
Please consider to delete information of “probiotics and prebiotics” in this part. If the authors would like to remain them in this content, information that these 2 things affect to PRRs in crustaceans, needs verification.
References
Please carefully check some minor errors in this part, especially for scientific names (both short and full forms), inconsistency of journal format, typos etc. These include references [36], [48], [52], [84], [86], [92], [108], [120], [121], [126], [128].
Author Response
Response to the decision letter
Dear editor,
We greatly appreciated the editor and the anonymous reviewers for critical reading of the manuscript and giving so many helpful and constructive advices. The manuscript has been carefully revised and greatly improved for related information according to the comments and suggestions of the editor and the reviewers. We indicated the changes in the revised manuscript to highlight in yellow.
The point-to-point answers to the comments of the reviewers are as follows:
To Reviewer #1:
The review manuscript entitled “Role of Cell-Surface Receptors in the Innate Immune System of 2 Crustaceans in Response to White Spot Syndrome Virus” by Tran et al., describing a vigorously scientific attempts to narrate functional roles and expression analysis of pattern recognition receptor genes crucial taking part in response to white spot syndrome virus, which is a major pathogen that causes severe diseases (white spot disease; WSD) in several crustaceans in decapod invertebrates, especially for prawn, shrimp and crab.
All in all, the review manuscript is well-written and well describe critical information useful for understanding immune pathways in crustaceans used to respond to WSSV especially for host-pathogen interaction via immune receptor molecules of crustaceans. However, it still remains some minor concerns that the authors should revise and provide some correction to improve the quality of the review manuscript, these included;
Reply: Many thanks for your comments.
Abstract
Line 11. Correct “white spot syndrome” to “white spot diseases (WSD)” and in every place.
Reply: The information has been corrected in the revised manuscript.
- Introduction
Line 29. To increase the better flow of the sentence, please correct “in invertebrates” to “in crustaceans”
Reply: The information has been replaced in the revised manuscript.
Line 33-35. The logic of the following is awkward; “Cell-surface receptors (including pattern recognition receptors-PRRs) play an important role in the innate immune…”, since many cell-surface receptors do not play roles in innate immune system…”, please properly revise.
Reply: The sentence has been changed in the revised manuscript.
Line 76. Please carefully consider about “in cultured aquatic animals”, more specific terms should replace “aquatic animals”, since only crustaceans are normally affected by WSSV.
Reply: The term “aquatic animals” has been replaced with “crustaceans”.
Basic interaction of PAMPs and PRRs should be properly raised in this part to indicate the major direction of this review.
- Toll receptors
Line 82 and line 103. Please keep consistent either using “white leg shrimp (Litopenaeus vannamei) or Pacific white shrimp (Penaeus vannamei)”, since it is the same organisms.
Reply: The species has been changed into white leg shrimp.
Line 116. Change a grammatical error “has” to “have”.
Reply: The word has been changed
Line 159-162. Please recheck the truth of the following sentence; “RNAi knockdown of LvIKKβ and LvIKKε significantly reduced the expression of genes related to the NF-κB signaling pathway, such as LvPEN2, LvPEN3, LvPEN4, Lvlysozyme, Lvcrustin1, and Lvcrustin2, as well as increased the survival rates of shrimp after WSSV infection [46].”. How reduction of such “LvPEN2, LvPEN3, LvPEN4, Lvlysozyme, Lvcrustin1, and Lvcrustin2” increase survival rate of shrimp after WSSV infection??
Reply: The information shown in the manuscript is true evidence, which was explained by the following sentence “The increased resistance to WSSV in LvIKKβ-or LvIKKε-silenced shrimp is possible by the requirement of IKK-NF-κB signaling pathway activation for WSSV infection [46]” that has been added in the revised manuscript.
- C-type lectins
Line 216-217. Please consider the following sentence carefully; “FcLec3 has been characterized to be a receptor located on the surface of shrimp cells that is able to recognize VP28 using pull-down assay [22].”. The authors should specify “shrimp cells”. Does FcLec3 ban be ubiquitously observed in every kind of cells in shrimp??
Reply: The shrimp cells (F cells in hepatopancreas) has been added to the revised manuscript; actually, in this study, the authors (Wang et al., 2009 [22]) confirmed that FcLec3 is only distributed in the F cells in hepatopancreas.
Line 235. Correct “Collective”.
Reply: The word has been changed into “Collectively”
Line 243. Please be careful indication of “gene or protein” status using Italic forms in all gene status in everywhere else.
Reply: The comment is noticed, and the font types indicating “gene or protein” have been confirmed in the revised manuscript.
Line 252-253. Please correct grammatical errors and direction of meaning of the following sentence; “The evidence for a regulatory role of CTL in the association with the activation of many signaling pathways in response to WSSV infection in crustaceans.”
Reply: Thank you very much for your meticulous reading of the manuscript, the sentence has been changed in the revised manuscript.
- Scavenger receptors
Line 260-261. Please carefully consider the following sentence; “SRs play a key role in the innate immunity of both mammals and invertebrates through recognizing pathogen-associated molecular patterns (PAMPs) and in”. To increase the better flow, please change “mammals” to “vertebrates”.
Reply: The “mammals” has been replaced with “vertebrates” in the revised manuscript.
Line 287-289. Please carefully check the following information; “Sp-SRB was induced in both hemocytes and hepatopancreas of mud crabs after challenge with WSSV or other pathogens (including V. parahaemolyticus, LPS, or PolyI:C) [74].”. LPS or POlyI:C is not classified as pathogens.
Reply: The sentence has been changed into “Sp-SRB was induced in both hemocytes and hepatopancreas of mud crabs after challenge with either WSSV, V. parahaemolyticus, LPS, or PolyI:C [74]”
Line 291. Please correct grammatical errors of relative pronounce “which” and many places throughout the manuscript.
Reply: These grammatical errors have been corrected.
Line 296. Correct “Casepase-3” to “casepase-3”.
Reply: The word has been changed.
- b-intergrin
Line 322. The following information is vague; “Surprisingly, integrin has been reported to be a heterodimeric protein including α and β subunits, but α subunit has not yet been found in 323 crustaceans [79]”. The authors should specify, what kind of animals that heterodimeric intergrin α and β subunits is reported.
Reply: The information has been added in the revised manuscript.
- Globular C1q receptor
Line 434. Please check what kind of cells used for this experiment, “ a hematopoietic tissue cell culture or a primary cell-line)”??
Reply: Thanks for your comment, the authors have checked and confirmed the cells used for this experiment are hematopoietic tissue cells.
- Lipopolysaccharide-and β-1,3-glucan-binding protein
Line 460. Please verify and clarify “the vaccinated group”??
Reply: The vaccinated group herein mentioned the “Bac-VP28-treated shrimp”, to make it clearly, the authors have used the term “Bac-VP28-treated shrimp” in the revised manuscript.
- Chitin-binding protein
Line 495. Please check “chitin” in this part. Does it refer to “chitin or chitin binding protein”??
Reply: The authors have checked and confirmed it indicates “chitin”.
Line 511. Correct “suggests”.
Reply: The word has been corrected
- Conclusion
Please consider to delete information of “probiotics and prebiotics” in this part. If the authors would like to remain them in this content, information that these 2 things affect to PRRs in crustaceans, needs verification.
Reply: Thank you for your suggestions, we have deleted the relating information.
References
Please carefully check some minor errors in this part, especially for scientific names (both short and full forms), inconsistency of journal format, typos etc. These include references [36], [48], [52], [84], [86], [92], [108], [120], [121], [126], [128].
Reply: The information has been checked and corrected appropriately. However, in some references (for example, [120] Sritunyalucksana et al. PmRab7 is a VP28-binding protein involved in white spot syndrome virus infection in shrimp. J. Virol. 2006, 80, 10734-10742) did not provide the scientific name of the species in the title of the paper, so we can add them.
Yours Sincerely,
Shengkang Li
Institute of Marine Sciences
Shantou University

Reviewer 2 Report
The manuscript reviewed the reported cell-surface receptors in responsive to WSSV infection in crustaceans. The authors provided a comprehensive and clear content, which is readable and helpful for readers. A major concern from the reviewer is that the definition of cell-surface receptors is not exactly consistent with the content of the manuscript. Usually, cell-surface receptors are considered as transmembrane receptors, which locate on cell-surface, receive signals from outside the cell and transduce them inside the cell. Several kinds of molecules reviewed in the manuscript do not fit the features. Most C-type lectins are soluble proteins and only few ones have transmembrane motif. As written in the manuscript, the crustacean C1qR is characterized to contain mitochondrial targeting sequence and a mitochondrial acidic matrix protein (MAM33) domain. Although a previous study detected positive signals of a chitin-binding protein on the hemocyte surface, CBPs are regards as extracellular molecules. Ras-associated binding proteins are mainly intracellular proteins. Although these proteins were reported direct interacting with WSSV envelope proteins, it is not quite reasonable to deem them as cell-surface receptors. In order to present the reader with a complete picture of the direct interaction between WSSV envelope proteins and host proteins, it is reasonable to replace the phrase cell-surface receptors with another one. The content of the Introduction also needs to be adjusted accordingly.
Author Response
Response to the decision letter
Dear editor,
We greatly appreciated the editor and the anonymous reviewers for critical reading of the manuscript and giving so many helpful and constructive advices. The manuscript has been carefully revised and greatly improved for related information according to the comments and suggestions of the editor and the reviewers. We indicated the changes in the revised manuscript to highlight in yellow.
The point-to-point answers to the comments of the reviewers are as follows:
To Reviewer #2:
The manuscript reviewed the reported cell-surface receptors in responsive to WSSV infection in crustaceans. The authors provided a comprehensive and clear content, which is readable and helpful for readers. A major concern from the reviewer is that the definition of cell-surface receptors is not exactly consistent with the content of the manuscript. Usually, cell-surface receptors are considered as transmembrane receptors, which locate on cell-surface, receive signals from outside the cell and transduce them inside the cell. Several kinds of molecules reviewed in the manuscript do not fit the features. Most C-type lectins are soluble proteins and only few ones have transmembrane motif. As written in the manuscript, the crustacean C1qR is characterized to contain mitochondrial targeting sequence and a mitochondrial acidic matrix protein (MAM33) domain. Although a previous study detected positive signals of a chitin-binding protein on the hemocyte surface, CBPs are regards as extracellular molecules. Ras-associated binding proteins are mainly intracellular proteins. Although these proteins were reported direct interacting with WSSV envelope proteins, it is not quite reasonable to deem them as cell-surface receptors. In order to present the reader with a complete picture of the direct interaction between WSSV envelope proteins and host proteins, it is reasonable to replace the phrase cell-surface receptors with another one. The content of the Introduction also needs to be adjusted accordingly.
Reply: Thank you very much for your comments, we have changed the information appropriately in the revised manuscript.
Yours Sincerely,
Shengkang Li
Institute of Marine Sciences
Shantou University
